# Curcumin-Loaded Microspheres Are Effective in Preventing Oxidative Stress and Intestinal Inflammatory Abnormalities in Experimental Ulcerative Colitis in Rats

**DOI:** 10.3390/molecules27175680

**Published:** 2022-09-02

**Authors:** Dana Hales, Dana-Maria Muntean, Maria Adriana Neag, Béla Kiss, Maria-Georgia Ștefan, Lucia Ruxandra Tefas, Ioan Tomuță, Alina Sesărman, Ioana-Adela Rațiu, Alina Porfire

**Affiliations:** 1Department of Pharmaceutical Technology and Biopharmacy, Faculty of Pharmacy, University of Medicine and Pharmacy “Iuliu Haţieganu”, 41 Victor Babeș Street, 400012 Cluj-Napoca, Romania; 2Department of Pharmacology, Toxicology and Clinical Pharmacology, Faculty of Medicine, University of Medicine and Pharmacy “Iuliu Haţieganu”, 400337 Cluj-Napoca, Romania; 3Department of Toxicology, Faculty of Pharmacy, University of Medicine and Pharmacy “Iuliu Haţieganu”, 400012 Cluj-Napoca, Romania; 4Department of Molecular Biology and Biotechnology, Centre for Systems Biology, Biodiversity and Bioresources (3B), Faculty of Biology and Geology, Babes-Bolyai University, 5-7 Clinicilor Street, 400006 Cluj-Napoca, Romania; 5Molecular Biology Center, Institute for Interdisciplinary Research in Bio-Nano-Sciences, Babes-Bolyai University, 42 Treboniu Laurian Street, 400271 Cluj-Napoca, Romania; 6Department of Nephrology, Faculty of Medicine and Pharmacy, University of Oradea, 410073 Oradea, Romania

**Keywords:** ulcerative colitis, oxidative stress, inflammation, Eudragit^®^ FS, colon-specific delivery, microparticles, curcumin

## Abstract

Curcumin’s role in the treatment of ulcerative colitis (UC) has been proven by numerous studies, but its preventive administration, with the aim of reducing the remission episodes that are characteristic of this disease, must be further investigated. This study investigates the effects of a novel curcumin-loaded polymeric microparticulate oral-drug-delivery system for colon targeting (Col-CUR-MPs) in an experimental model of UC. Male Wistar rats (n = 40) were divided into five groups (n = 8), which were treated daily by oral gavage for seven days with a 2% aqueous solution of carboxymethylcellulose sodium salt (CMCNa) (healthy and disease control), free curcumin powder (reference), Col-CUR-MPs (test) and prednisolone (reference) prior to UC induction by the intrarectal administration of acetic acid (AA), followed by animal sacrification and blood and colonic samples’ collection on the eighth day. Col-CUR-MPs exhibited an important preventive effect in the severity degree of oxidative stress that resulted following AA intrarectal administration, which was proved by the highest catalase (CAT) and total antioxidant capacity (TAC) levels and the lowest nitrites/nitrates (NOx), total oxidative status (TOS) and oxidative stress index (OSI) levels. Biochemical parameter analysis was supported by histopathological assessment, confirming the significant anti-inflammatory and antioxidant effects of this novel colon-specific delivery system in AA-induced rat models of UC. Thus, this study offers encouraging perspectives regarding the preventive administration of curcumin in the form of a drug delivery system for colon targeting.

## 1. Introduction

Ulcerative colitis (UC), one of the two forms of inflammatory bowel disease (IBD), is a chronic inflammatory condition that affects millions of people around the world [1,2]. Even though the pathogenesis and treatment of the disease has already been studied for decades, there are no standard or definitive treatments for this disease. Therefore, UC persists throughout the patient’s whole life, with the clinical course being characterized by alternating periods of exacerbation and remission. The affected regions are the mucosa and submucosa, mainly in the colon and rectum. The etiology and pathogenesis of IBD are not fully understood, however considerable progress is constantly being made in this direction. So far, it is known that IBD is an immune-mediated condition, which appears to be determined by the complex interaction between several types of factors (e.g., environmental, microbial, genetic and immunological factors), and which occurs in genetically susceptible individuals [1,2,3]. These interactions induce an excessive immune activity of lymphocytes and an increased production of inflammatory cytokines (e.g., TNF-α, IL-6, IL-1β, IL-17, etc.), disturbing the homeostasis of the intestinal mucosa and leading to chronically active inflammation, which is closely linked to the excessive production of reactive oxygen species (ROS) [4,5,6,7].

Therefore, oxidative stress, the imbalance between ROS production and elimination, appears to have an important influence on the development and progression of IBD [4]. The substantial antioxidant system, which is found in a normal intestinal epithelium, having the role of withstanding oxidative damage, is usually present in low levels in the UC colonic tissues, due to disturbed gut homeostasis. On the contrary, the overexpression of antioxidant enzymes determines the reduction of inflammation in UC [8]. Therefore, the measurement of oxidative stress biomarkers in blood or tissue samples has been reported to provide information on the severity of the disease or the antioxidant capacity of a studied therapeutic agent. Among the most relevant biomarkers used for oxidative stress assessment are catalase (CAT), total antioxidant capacity (TAC), nitrites and nitrates (NOx), total oxidant status (TOS), malondialdehyde (MDA) and the oxidative stress index (OSI) [9,10,11].

The first-line drug categories in the treatment of IBD are usually aminosalicylates, while corticosteroids and immunosuppressants are indicated in advanced cases [1,12]. Monoclonal antibodies are used as an alternative in patients with chronic inflammatory diseases that do not respond to the initial treatment [7,13,14]. The disadvantage of the commercially available treatments is the occurrence of serious side effects, which cannot be avoided or maintained at a minimum level. Therefore, there is an acute need to develop specific therapeutic approaches that are capable of offering high therapeutic efficacy and reduced toxicity [1]. For this reason, recent trends in the research of IBD-related treatment strategies are oriented towards studying chemopreventive agents in order to establish their potential for reducing disease progression. Several medicinal herbs were studied for the treatment of UC due to their content of chemopreventive agents, which are phytochemicals derived from fruits and vegetables, such as resveratrol, capsaicin, genistein or curcumin [1,15].

*Curcuma longa* is a herbal product that is commonly used as a spice or food additive, but has been also exploited since ancient times in traditional medicine for its multiple beneficial effects, such as antioxidant, anti-inflammatory, anti-tumor, anti-bacterial, wound healing and hypocholesterolemic activities [16]. The rhizome of this plant is used as a powder, called turmeric, which contains at least three curcuminoids: bisdemethoxycurcumin, demethoxycurcumin and curcumin. Curcumin, the most abundant of the three polyphenols, has been used for medical applications, such as the treatment of inflammation and pain [17,18,19]. In IBD, curcumin appears to exert a prophylactic role due to its ability to adjust the oxidant/antioxidant balance and modulate the release of inflammatory mediators, such as TNF-α and nitric oxide (NO) [12]. Regarding its toxicity, various clinical studies have demonstrated the safety of curcumin for humans, even when administered in high doses [20,21,22]. Despite the numerous beneficial aspects, curcumin has a low solubility in water, therefore it is poorly absorbed from the gastrointestinal tract, it is unstable at neutral and alkaline pH and is rapidly metabolized and eliminated, leading to poor oral bioavailability [12,18,23]. Oral administration is preferred in IBD, since the systemic toxicity is reduced and the efficacy of therapeutic agents is higher. However, colon-specific drug delivery is a challenging task, as certain drugs are degraded in acidic pH, inactivated by digestive enzymes or simply encounter difficulties in being absorbed by the colonic mucosa [24]. In order to overcome these drawbacks and improve the drug’s local delivery, curcumin has been encapsulated in different colon-specific carriers (e.g., pellets, micro- and nanoparticles, liposomes or micelles) capable of solubility improvement, drug degradation prevention and direct delivery to colitis tissues [2,12,21,25].

Our previous study focused on the development of a novel curcumin-loaded polymeric microparticulate oral-drug-delivery system (Col-CUR-MPs) for colon targeting, by a quality-by-design approach (QbD) [26]. Of the 17 formulations of the experimental design, the formulation which contained 40 mg/mL of a mixture of polycaprolactone and Eudragit^®^ FS (50:50, *w*/*w*) and 2 mg/mL curcumin was chosen to be used in the present in vivo study. The characteristic that was decisive in choosing this formulation for the in vivo administration was the fact that the percentage of curcumin released after 2 h in an acidic environment was the lowest (14.50% ± 2.0%), and closest to the 10% limit required for colonic-release formulations.

The aim of this study was to investigate whether the Col-CUR-MPs are able to specifically target the colon and attenuate oxidative stress and inflammation in an in vivo experimental model of acetic-acid (AA)-induced UC. Although most colitis models evaluate the treatment of already established ulcerative colitis with different therapeutic agents, there are a few studies that have dealt with the evaluation of the possibility of preventing tissue damage and exacerbation of the disease through the preventive administration of protective agents. These studies have demonstrated the preventive effect of the studied treatments on colitis and their ability to lower inflammation and to reduce the damage to the colonic mucosa [27,28].

Thus, our study sought to prove the beneficial effects of the preventive administration of Col-CUR-MPs, prior to UC induction, as a background medication which would reduce disease exacerbations. In this regard, we evaluated the efficacy of the developed system against an AA-induced rat model of UC by measuring the plasma or serum levels of several oxidative stress biomarkers and evaluating the colonic tissues of the experimental animals.

## 2. Results

### 2.1. Evaluation of Plasma Antioxidant Response

#### 2.1.1. Catalase (CAT)

Plasma CAT levels reported in our study, presented in Figure 1a, were decreased in the AA group (67.7 ± 19.1 U/mL) compared with the control group (70.9 ± 16.7 U/mL), which is a clear indicator of oxidative stress in the colon tissue of the AA group. CAT levels in the AA_CCS and AA_PLP groups showed similar values (53.5 ± 13.5 U/mL and 44.2 ± 9.5 U/mL, respectively) and were lower than the levels in the AA group. Regarding the AA_CCMP group, the results showed the highest CAT levels (98.1 ± 27.7 U/mL). However, the results obtained for the treated groups displayed no statistically significant difference as compared with the AA group (*p* > 0.05).

#### 2.1.2. Total Antioxidant Capacity (TAC)

The antioxidant ability of the tested colonic drug delivery system was explored by measuring the TAC in plasma (Figure 1b). For the AA group, the TAC was significantly lower than that of the control group (*p* = 0.02), showing the important decline in the antioxidant defense caused by colitis induction. Among the groups which received treatment/antioxidant agents, the highest TAC was observed in the AA_CCMP group (0.19 ± 0.03 mmol Trolox equivalent/L), indicating an important antioxidant capacity provided by the Col-CUR-MPs system (*p* = 0.02, AA_CCMP vs. AA). Furthermore, the levels of the TAC obtained for the AA_CCS and AA_PLP groups were not significantly different from the TAC level of the AA group (p value was higher than 0.05), however the AA_PLP group showed a mean TAC level (0.18 ± 0.07 mmol Trolox equivalent/L) comparable to that of the AA_CCMP group.

### 2.2. Evaluation of Plasma Oxidative Stress

#### 2.2.1. Total NO Metabolites (Nitrites and Nitrates, NOx)

Total NO metabolites were measured in serum samples in order to evaluate the nitrosative stress. In the AA group, a significant increase of NOx level was observed (770.2 ± 218 μmol/L, *p* < 0.05 vs. control), signaling the important nitrosative stress that accompanies the inflammation caused by AA administration. The AA_CCMP group stood out by showing the lowest level of NOx, 521.6 ± 42 μmol/L, which was significantly lower (*p* < 0.05) than that of the AA group (Figure 2a), and was closely followed by the AA_CCS group (528.6 ± 125 μmol/L, *p* < 0.05, AA_CCS vs. AA group). The AA_PLP group showed a lower level of NOx than the AA group (657.2 ± 173 μmol/L), but the difference between the two groups was not statistically significant.

#### 2.2.2. Total Oxidant Status (TOS)

The TOS level increased greatly in the AA group (136.7 ± 25 μmol H_2_O_2_ equivalent/L) compared with the control group (69.3 ± 21 μmol H_2_O_2_ equivalent/L), with the difference between the two groups being statistically significant (*p* < 0.0001), and indicating a high pro-oxidative status due to the induced inflammation. Serum TOS values were significantly lower (*p* < 0.01 for both) in the AA_CCS and AA_CCMP groups compared with the AA group (77.4 ± 13 μmol H_2_O_2_ equivalent/L and 69.7 ± 23 μmol H_2_O_2_ equivalent/L, respectively), confirming that free curcumin and especially Col-CUR-MPs offer an important antioxidant protection. The AA_PLP group showed the lowest TOS levels (44.8 ± 13 μmol H_2_O_2_ equivalent/L) compared with the AA group (*p* < 0.0001), and thus, the highest antioxidant capacity (Figure 2b).

#### 2.2.3. Lipid Peroxidation (Malondialdehyde, MDA)

The MDA was measured in plasma, as an indicator of lipid peroxidation, and the results were illustrated in Figure 2c. The MDA level was considerably increased (*p* < 0.05) in the AA group (3.75 ± 0.7 nmol/mL) compared with the control group (2.95 ± 0.2 nmol/mL). For the groups treated with free curcumin, Col-CUR-MPs and prednisolone, there was a slight decrease in the mean MDA levels (3.48 ± 0.5 nmol/mL, 3.51 ± 0.5 nmol/mL and 3.52 ± 0.5 nmol/mL, respectively) compared with the AA group, but the differences between te hMDA plasmatic levels of the AA group and of the treated groups were not statistically significant.

### 2.3. Evaluation of Oxidative Stress Index (OSI)

The oxidative status presented in Figure 3 revealed an important increase of the OSI for the AA group (0.91 ± 0.23, *p* < 0.0001 vs. the control group), indicating a significant degree of inflammation-associated oxidative stress in the untreated colitis group. For all the treated groups, there was a reduction of the OSI values compared with the AA group, and, notably, the level of significance was important for the AA_CCMP group (0.35 ± 0.11, *p* < 0.0001 vs. the AA group) and for the AA_PLP group (0.27 ± 0.06, *p* < 0.0001 vs. the AA group).

### 2.4. Histopathological Assessment

As shown in Figure 4a,b, the colonic tissue of the control group presented a normal structure of the components of all intestinal tunics, without degenerative or alterative changes. In the AA group (Figure 4c,d), the tissue showed a generalized hemorrhagic necrosis and pronounced edema in all tunics of the intestine (mucosa, submucosa and muscle). The AA_PLP group (Figure 4i,j) presented zonal hemorrhagic necrosis, mild congestion and discrete submucosal edema. The histopathological changes observed for the AA_CCS group (Figure 4e,f) were similar to the AA_PLP group, the hemorrhages and edema in the submucosa being, however, slightly more intense in the AA_CCS group; the functionality of the organ was affected by a proportion of about 60–70%. Remarkably, in the AA_CCMP group (Figure 4g,h), the hemorrhagic-necrotic colitis was superficial and affected only small parts of the tissue, the functionality of the organ being affected by a proportion of a maximum of 20%.

## 3. Discussion

There is a great need for efficient therapeutic agents in the treatment of IBD that are able to reduce the inflammation that is the cause of the disease symptoms, but also to determine as few side effects as possible. Moreover, there is also a need to understand the efficacy of preventive approaches in modifying the clinical course of IBD. Silveira et al. showed that the administration of a high-fiber diet prior to UC induction protected from acute colitis [29], therefore similar effects could be proven for other supplements or therapeutic agents of interest in IBD evolution.

Natural herbal products, such as curcumin, have caught increasing attention over the past several decades, therefore there are a large number of studies involving curcumin and pharmaceutical systems encapsulating curcumin. However, the ability of the colon drug delivery systems to transport the active pharmaceutical ingredients (APIs) at the specific site of action is an objective that is hard to achieve.

Due to the necessity of designing efficient colon drug delivery systems in the treatment of UC, we first conducted a study for the development of colon-targeted curcumin-loaded polymeric microparticles (Col-CUR-MPs) applying the QbD approach. The formulation that was chosen to be administered to animals in the in vivo experiment presented good physico-chemical characteristics. The particle size was 146.90 μm and the particle size distribution was 52.10%, representing an approximate average of the 17 formulations of the experimental plan. The drug loading (4.09%) and the entrapment efficiency (85.15%) were satisfactory, showing elevated levels. The high level of the manufacturing method yield (92.82%) showed the efficiency of the preparation process. Regarding the percentages of curcumin released, after 2 h in an acidic environment and 24 h in an intestinal-simulated environment, the Col-CUR-MPs released 14.50% ± 2.0% and 49.92% ± 18.7%, respectively [26]. The formulation chosen for the in vivo study was closest to the maximum 10% limit of drug-release in an acidic environment after 2 h, which is the most important condition for colonic-release formulations [30].

The encapsulated curcumin (Col-CUR-MPs) was administered preventively for seven days in parallel with free curcumin and prednisolone, before colitis induction, in order to establish the antioxidant and anti-inflammatory capacity of encapsulated curcumin in comparison with non-encapsulated curcumin and with one of the IBD reference treatments. The measurement of oxidative stress biomarkers and histopathological investigation showed that, overall, the encapsulated curcumin presented higher antioxidant and anti-inflammatory activities than free curcumin and prednisolone.

Regarding the disease control group, the intrarectal administration of AA to experimental animals induced a significant inflammatory response in plasma and serum, as highlighted by the reduced levels of CAT and TAC, and the increase of oxidative and nitrosative stress biomarkers (NOx, TOS, MDA and OSI).

CAT, one of the catalytic enzymes that has a key role as an enzymatic antioxidant in the defense mechanism against oxidative stress, is able to prevent cellular oxidative damage by degrading hydrogen peroxide (H_2_O_2_) into water and oxygen [31,32]. Low levels of CAT have been also been reported in other studies that researched rat or mouse UC models, confirming that CAT deficiency or malfunctioning is associated with many diseases whose evolution is favored by the appearance of ROS [33,34].

Regarding TAC, which shows the antioxidant response against free radicals produced in certain ROS-generating diseases, it has been previously reported that diminished TAC levels are associated with active UC [9,35,36]. Therefore, the results obtained in our study indicated an important oxidative stress caused by colonic tissue inflammation in the AA group.

On the contrary, oxidative and nitrosative stress biomarkers are normally increased in the disease group, as confirmed by our results. NO metabolites, nitrites and nitrates, were found in high levels in the plasma, feces and colonic lumen of patients with IBD, revealing elevated nitrosative stress [9,37].

TOS was reported to present increased levels following AA or other inflammatory agents’ administration. High TOS levels, such as the ones obtained in the disease group, indicate an enhancement of the total concentration of different oxidant molecules and thus, a marked pro-oxidative status [38,39,40,41].

MDA elevated levels are linked to oxidative stress and inflammatory conditions [42,43], such as IBD [34]. In our study, the intrarectal administration of AA resulted in a significant increase in MDA level compared with the control group, suggesting an important increase of lipid peroxidation in the colon tissue.

OSI is defined as the ratio of the TOS level to TAC level, therefore its magnitude is used to assess the degree of oxidative stress of inflammatory disorders and potentially to offer a diagnostic method in oxidative stress-determined diseases [11]. After AA intrarectal administration, there was a significant increase of OSI compared with the control group, which demonstrated the pro-oxidative status of the blood samples in the AA group [11,44].

The histopathological evaluation revealed clear colon tissue damage following AA administration (Figure 4c,d), the organ function being affected by more than 90%, compared with the healthy control group (Figure 4a,b), for which there were no degenerative or alterative changes in the colonic mucosa and submucosa of the experimental animals. More precisely, in the AA group one could observe generalized necrotic-hemorrhagic colitis, stasis, hemorrhages and marked edema.

The main purpose of our study was to investigate whether the preventive administration of encapsulated curcumin in induced UC revealed beneficial effects in the evolution of AA-induced acute inflammation and oxidant status. In brief, Col-CUR-MPs, free curcumin and prednisolone were administered to rats daily by oral gavage for seven consecutive days, before the induction of UC by the intrarectal administration of AA, followed by animal sacrification and blood and colonic samples collection on the eighth day.

Preventive administration of curcumin in patients with ulcerative colitis could be used as background therapy, which would reduce the exacerbations of the disease, while the encapsulation of curcumin in microparticles would ensure the specific transport to the inflamed colon-tissue target site. Most of the UC animal models reported in published articles focus on the treatment of previously induced colitis with various therapeutic agents, including curcumin [17,20,21]. There are only a few studies that evaluate the effect of prevention on the course of UC, of which even fewer highlight the antioxidant effects [27,28,29].

In our study, Col-CUR-MPs showed intestinal antioxidant effects in AA-induced UC through the marked increase of CAT and TAC levels, and the decrease of NOx, TOS, MDA and OSI levels.

Among the treated groups, the AA_CCMP group presented the highest CAT and TAC levels, therefore indicating a protective effect of Col-CUR-MPs on the colonic tissue and the reduction of oxidative stress, compared with the AA_CCS and AA_PLP groups (Figure 1). This confirmed the fact that the encapsulation of curcumin in polymeric microparticles is an efficient approach to tackling colonic oxidative stress. The colon-targeted drug delivery system was able to exert a protective effect, while the non-encapsulated form was inefficient in preventing CAT and TAC decline. Furthermore, even though prednisolone is a standard therapy in IBD, it was not capable of generating a comparable effect with the one of the Col-CUR-MPs regarding CAT, while the TAC level was comparable, but still inferior [27,45].

Additionally, the nitrosative stress was the lowest for the AA_CCMP group followed closely by the AA_CCS group (Figure 2a). This indicated the antioxidant potential of curcumin and especially the Col-CUR-MPs system in AA-induced UC, considering that there is a direct relationship between NO and ROS generation [46]. These results proved once more the superiority of the encapsulated curcumin compared to its free form, and to prednisolone.

Regarding TOS serum levels and OSI, non-encapsulated curcumin administration did not prevent the increase of OSI in a significant manner. However, when encapsulated in microparticles, curcumin demonstrated a clear capacity to limit OSI growth, comparable to that of prednisolone. TOS levels were similar for the AA_CCS and AA_CCMP groups, even though they were lower for the latter, and significantly reduced for the AA_PLP group. Our results are indicative of a good ability of Col-CUR-MPs to prevent the oxidative stress induced by AA administration and of their potential to protect colon tissues from inflammatory damage, similar to other research studies that reported a positive correlation between the administration of antioxidant agents and the reduction of TOS and OSI values [10,11,39,44]. The higher TOS value of the AA_CCMP group compared with the group that received prednisolone may be due to the short period of prevention that could be insufficient for an adequate amount of phytochemical agent to be accumulated in the colon tissue so as to diminish ROS and reduce oxidation [47]. In addition, a greater dose of curcumin could have led to superior antioxidant properties compared with the same dose of prednisolone.

MDA levels, indicating lipid peroxidation as ROS/RNS-related damage, are usually increased in groups with induced inflammation, but if anti-inflammatory or antioxidant agents are administered, MDA levels are significantly decreased [27,48,49]. We observed that the administration of curcumin, either in free or encapsulated form, and prednisolone was able to slightly prevent the increase of lipid peroxidation, however the results were not statistically significant (Figure 2c).

In the histopathological evaluation of colonic tissues originating from the rats in the AA_CCS, AA_CCMP and AA_PLP groups, beneficial effects were observed compared with the disease control group (Figure 4e–j). Our results showed that the animals treated with free curcumin (AA_CCS group) and prednisolone (AA_PLP group) were similar in terms of tissue damage, with the functionality of the organ being seriously affected, in a proportion of about 60-70% and 50%, in the AA_CCS group and the AA_PLP group, respectively. Both groups showed zonal necrotic-hemorrhagic colitis, however the AA_CCS group presented hemorrhages and submucosal edema, while in the AA_PLP group only mild congestion and submucosal discrete edema were observed. The lowest tissue damage was observed for the group treated with curcumin-loaded microparticles, more precisely colon-tissue function was affected to the lowest extent, while the hemorrhagic-necrotic colitis affected only small parts of the tissue and was superficial. This suggests that the encapsulation of curcumin in colon drug delivery microparticulate systems offers better protective effects in UC.

All the previously discussed results that indicated the superiority of the colon drug delivery system over non-encapsulated curcumin and prednisolone may support the hypothesis that the Col-CUR-MPs have a good capacity to prevent the release of the encapsulated drug in the upper parts of the gastrointestinal tract and to ensure its specific release in the colon, thus allowing drug accumulation at the target site and preventing oxidative stress and inflammation [26].

## 4. Materials and Methods

### 4.1. Reagents

The chemicals used for the preparation of microspheres were the following: curcumin, polycaprolactone (average mol wt 45,000) and poly(vinyl alcohol) (87–90% hydrolyzed, average mol wt 30,000–70,000) (Sigma-Aldrich, St. Louis, MO, USA), Eudragit^®^ FS 100 (Evonik Nutrition&Care GmbH, Darmstadt, Germany), dichloromethane, dimethyl sulfoxide, acetonitrile (Merck KGaA, Darmstadt, Germany) and ethyl acetate (Chimopar, Bucharest, Romania). The chemicals used for the in vivo animal experiment included prednisolone sodium phosphate (Henan Lihua Pharm. Co., Henan, China), carboxymethylcellulose sodium salt (Fluka Chemie GmbH, Buchs, Switzerland) and acetic acid (Sigma-Aldrich, St. Louis, MO, USA). All chemicals used for the biochemical parameters and histological evaluation were of analytical grade.

### 4.2. Preparation of Curcumin-Loaded Microspheres

An oil-in-water emulsion technique followed by solvent evaporation was used for the Col-CUR-MPs preparation (Figure 5). The preparation method and composition of the microspheres were optimized by the QbD approach in a previous study that aimed to develop colon-targeted curcumin-loaded polymeric microparticles using an Eudragit^®^ FS-polycaprolactone blend. Briefly, the organic phase consisted of a 2 mg/mL curcumin in dimethyl sulfoxide solution, mixed with an Eudragit^®^ FS 100 in ethyl acetate solution and polycaprolactone in dichloromethane solution, where the concentration of the polymer mixture was 40 mg/mL and the Eudragit^®^ FS proportion was 50%. In order to obtain the oil-in-water emulsion, the organic solution was added to a 20 mL aqueous solution of 2% poly(vinyl alcohol) and stirred at 340 rpm for 2 min using a magnetic stirrer. Next, the solvents were evaporated by adding the obtained emulsion to 100 mL double-distilled water and stirring at 510 rpm for 24 h with a magnetic stirrer, which resulted in the formation of solid microspheres due to the precipitation of the polymers. Finally, the microspheres were collected by filtration (nylon membrane, 0.45 μm) and dried at room temperature. The microspheres were characterized in terms of particle size, particle-size distribution, drug loading, entrapment efficiency, manufacturing method yield and percentages of curcumin released at different time intervals during 24 h in media that simulated the gastric and intestinal environments [26].

### 4.3. Animals and Experimental Protocol

The experimental protocol for the in vivo study was reviewed and approved by the Scientific Research Ethics Committee of the University of Medicine and Pharmacy “Iuliu Haţieganu” Cluj-Napoca (Decision no. 125/24.03.2020) and by the Sanitary Veterinary and Food Safety Authority in Cluj-Napoca (Decision no. 221/09.06.2020). The experiment was conducted in accordance with Directive 2010/63/EU regarding the protection of animals used for scientific purposes [50] and the Guide for the care and use of laboratory animals eighth edition [51], avoiding animals’ suffering. Forty Wistar male albino rats, weighing between 210 and 330 g, were supplied by the Experimental Medicine and Practical Skills Centre of the University of Medicine and Pharmacy “Iuliu Hațieganu” Cluj-Napoca, Romania. The animals were maintained under standard environmental conditions of temperature (22 ± 2 °C), humidity (45 ± 10%) and light (12 h /12 h light/dark cycle), and had access to standard pelleted feed and tap water ad libitum throughout the experiment.

The animals were randomly divided into five groups (n = 8) as follows: (1) normal control group (C), (2) disease control group or acetic acid group (AA), (3) curcumin group (AA_CCS), (4) Col-CUR-MPs group (AA_CCMP) and (5) prednisolone group (AA_PLP). Each group was treated daily by oral gavage for 7 consecutive days, as follows: groups 1 and 2 received a 2% aqueous solution of carboxymethylcellulose sodium salt (CMCNa), groups 3 and 4 received the same dose of curcumin (15 mg/kg), but under a different form-free (non-encapsulated) curcumin powder and Col-CUR-MPs, respectively, both suspended in a 2% aqueous solution of CMCNa, while group 5 received prednisolone sodium phosphate (2 mg/kg) dissolved in a 2% aqueous solution of CMCNa. After 7 days of treatment, 1.5 mL of distilled water was administered intrarectally to group 1, while groups 2–5 received 1.5 mL 4% AA solution for the induction of UC. On the eighth day, blood samples were collected through retroorbital punction and after animal sacrification, colonic tissue was collected. In order to evaluate the effects of Col-CUR-MPs administration in AA-induced colitis, oxidative and nitrosative stress assessment (CAT, TAC, NO metabolites, TOS, MDA), as well as histological assessment of colon tissue sections were performed.

### 4.4. Evaluation of Plasma Antioxidant Response

#### 4.4.1. Catalase (CAT)

CAT levels in plasma were assayed by a previously described method, developed in 1984 by Aebi [52]. The method is based on following the decomposition of H_2_O_2_ directly by measuring the decrease in absorbance at 240 nm. Thus, the plasma samples were mixed with H_2_O_2_, which started the reaction, and the decrease in absorbance was recorded at 240 nm for one minute using a spectrophotometer (Specord 250 Plus, Analytik Jena, Jena, Germany). CAT activity can be calculated by considering the difference in absorbance per unit time and knowing that a catalytic unit decomposes 1 μmol H_2_O_2_/minute, at pH = 7 and at 25 °C.

#### 4.4.2. Total Antioxidant Capacity (TAC)

The total antioxidant capacity (TAC) in plasma was measured by a Trolox Equivalent Antioxidant Capacity (TEAC) Assay, a spectrophotometric method developed by Erel [53], and applied by several groups of researchers [27,47,54]. The principle of the method relies on the ability of antioxidants to decolorize the blue-green 2,2′-azinobis-(3-ethylbenzothiazoline-6-sulfonic acid radical cation (ABTS(*+)), depending on their concentrations and antioxidant capacities. The measurement was performed by spectrophotometry at 660 nm (Specord 250 Plus, Analytik Jena, Jena, Germany) and the rate of bleaching was inversely related to the TAC of the sample. The calibration curve was constructed using Trolox, a water-soluble analogue of vitamin E used as a standard for measuring the antioxidant capacity of complex mixtures [55]. The results were expressed as mmol Trolox equivalent/L.

### 4.5. Evaluation of Plasma Oxidative Stress

#### 4.5.1. Total NO Metabolites (Nitrites and Nitrates, NOx)

The simultaneous evaluation of NO metabolites, nitrates and nitrites (NOx), was carried out according to the method developed by Miranda et al. [56]. The method is based on the reduction of nitrates to nitrites by vanadium (III), followed by transformation according to an acidic Griess reaction and detection by UV-visible spectroscopy at 540 nm (Specord 250 Plus, Analytik Jena, Jena, Germany) [44].

#### 4.5.2. Total Oxidant Status (TOS)

The total oxidation status (TOS) was measured by an automated, colorimetric method developed by Erel [38]. The principle of the assay is based on the oxidation of ferrous ion to ferric ion by several oxidants, in an acidic medium, followed by the measurement of ferric ion by xylenol orange. The results were expressed as mmol H_2_O_2_ equivalent/L [44].

#### 4.5.3. Lipid Peroxidation (Malondialdehyde, MDA)

In order to assess lipid peroxidation, a quantitative analysis of malondialdehyde (MDA) was performed. The plasma samples were hydrolyzed at 60 °C in a water bath, in the presence of NaOH, followed by protein precipitation with perchloric acid and derivatization with 2,4-dinitrophenylhydrazine. The derivatization product was extracted in n-hexane, followed by the evaporation of the organic layer under a stream of nitrogen and dissolution of the residue in the mobile phase. The samples prepared in the above-mentioned conditions were analyzed by UPLC-PDA, using a Waters Acquity UPLC system coupled with a Waters Acquity photo diode array detector (Waters, Milford, MA, USA) set at 301 nm and a BEH C18 analytical column (50 mm × 2.1 mm i.d., 1.7 mm) from Waters (Waters, Milford, MA, USA). Chromatographic separation (runtime of 7.5 min) was achieved using gradient elution and a mixture of 1% formic acid/acetonitrile as the mobile phase (flow rate: 0.3 mL/min) [44].

### 4.6. Evaluation of Oxidative Stress Index (OSI)

The oxidative stress index (OSI), an indicator of the degree of oxidative stress, was calculated as the percent ratio of the total oxidant status to the total antioxidant status [57,58]. The OSI value was calculated with the following formula: OSI (arbitrary unit, AU) = TOS (mmol H_2_O_2_ equivalent/L) / TAC (mmol Trolox equivalent/L) [44].

### 4.7. Histopathological Assessment

A histological investigation was performed using a previously described method [59]. Colon fragments were collected and fixed in 10% formalin for 3 days, then dehydrated with ethyl alcohol in increasing concentrations (70°, 96° and absolute alcohol), clarified with 1-butanol and embedded in paraffin. Cross tissue sections with a thickness of 5 μm were obtained, then stained using the Goldner’s trichrome method to observe colonic inflammation and damage. The obtained histological preparations were examined using an Olympus BX41 microscope (Tokyo, Japan) equipped with an Olympus E-330 (Tokyo, Japan) digital camera for capturing the images.

### 4.8. Statistical Analysis

All the results are expressed as the mean ± standard deviation (SD). The statistical analyses were performed using GraphPad Prism 8.4.3 statistical software (GraphPad, San Diego, CA, USA). A *p*-value of less than 0.05 was considered statistically significant.

## 5. Conclusions

In the present study, a novel curcumin-loaded polymeric microparticulate oral-drug-delivery system (Col-CUR-MPs) for colon targeting was administered in order to evaluate whether this system could be used for the prevention of AA-induced colitis. Compared to non-encapsulated curcumin, the curcumin-loaded microparticulate system demonstrated a much higher antioxidant and anti-inflammatory activity, presenting the most elevated levels of CAT and TAC, a lower oxidative stress for most of the determined parameters (NOx, TOS, OSI), and a significantly lower amount of tissue damage. In contrast with prednisolone, the drug delivery system for colon targeting exhibited a superior antioxidant capacity, yet the oxidative stress results were not all in favor of encapsulated curcumin, as it showed higher TOS and OSI levels.

Considering all the studied aspects, the data collected in this study provide valuable evidence that it is necessary for curcumin to be administered in a colonic-release carrier in order to have the desired effect, namely the specific delivery in the colon in order to ensure sufficient drug concentrations to combat oxidative stress and inflammation. Given that the encapsulated-curcumin-treated group performed particularly well compared with the free-curcumin-treated group and the reference treatment, offers encouraging insights.

However, there are a few major limitations in this study that could be addressed in future research. First, this research focused on the study of a unique formulation, so that carrying out a study that would include several formulations with different compositions, for example from the point of view of the type of polymers and their concentration, would provide more information in order to choose the most efficient system. Second, the system must be characterized deeper regarding its antioxidant and anti-inflammatory capacity, adding more oxidative stress and inflammation parameters for blood, but also for tissue samples, in order to have a more complex picture of the effects of such a system in the colitis-damaged tissue. Equally important, increasing the period of the preventive administration of microparticles or increasing the dose of administered curcumin could lead to valuable information.

In conclusion, this research presents new insights into the medicinal benefit of curcumin on its antioxidant and anti-inflammatory ability. Enriching the scientific data with information to prove the prevention of UC exacerbations by background administration of these microparticles could contribute to promoting curcumin as an alternative preventive measure against UC. Therefore, more in-depth studies are necessary in the future to shed light on the means through which the Col-CUR-MPs lead to reduced oxidative stress and inflammation and to try to explore and optimize this system as much as possible.

## Figures and Tables

**Figure 1 molecules-27-05680-f001:**
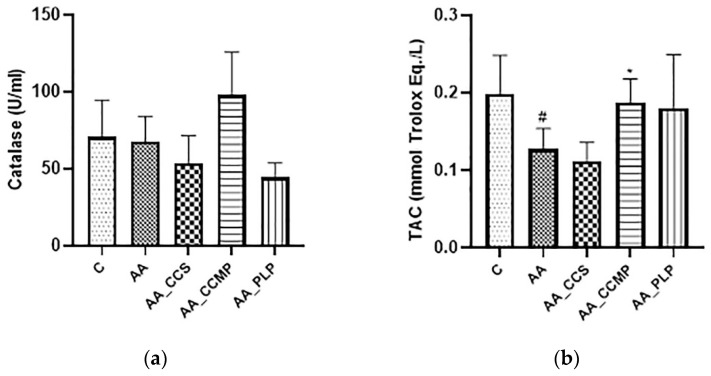
Catalase (CAT) (**a**) and total antioxidant capacity (TAC) (**b**) levels. Data are expressed as the mean ± SD for each group. # *p* < 0.05 indicates a significant difference in comparison with the control group; * *p* < 0.05 indicates a significant difference in comparison with the AA group. Abbreviations: C, normal control group; AA, acetic acid group; AA_CCS, curcumin group; AA_CCMP, Col-CUR-MPs group; and AA_PLP, prednisolone group. The bars represent mean values with standard deviations.

**Figure 2 molecules-27-05680-f002:**
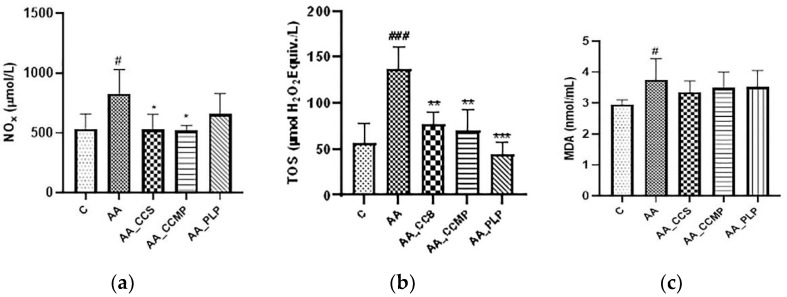
Nitrites and nitrates (NOx) (**a**), total oxidant status (TOS) (**b**) and malondialdehyde (MDA) (**c**) levels. Data are expressed as the mean ± SD for each group. # *p* < 0.05 and ### *p* < 0.0001 indicate a significant difference in comparison with the control group; * *p* < 0.05, ** *p* < 0.01 and *** *p* < 0.0001 indicate a significant difference in comparison with the AA group. Abbreviations: C, normal control group; AA, acetic acid group; AA_CCS, curcumin group; AA_CCMP, Col-CUR-MPs group; and AA_PLP, prednisolone group. The bars represent mean values with standard deviations.

**Figure 3 molecules-27-05680-f003:**
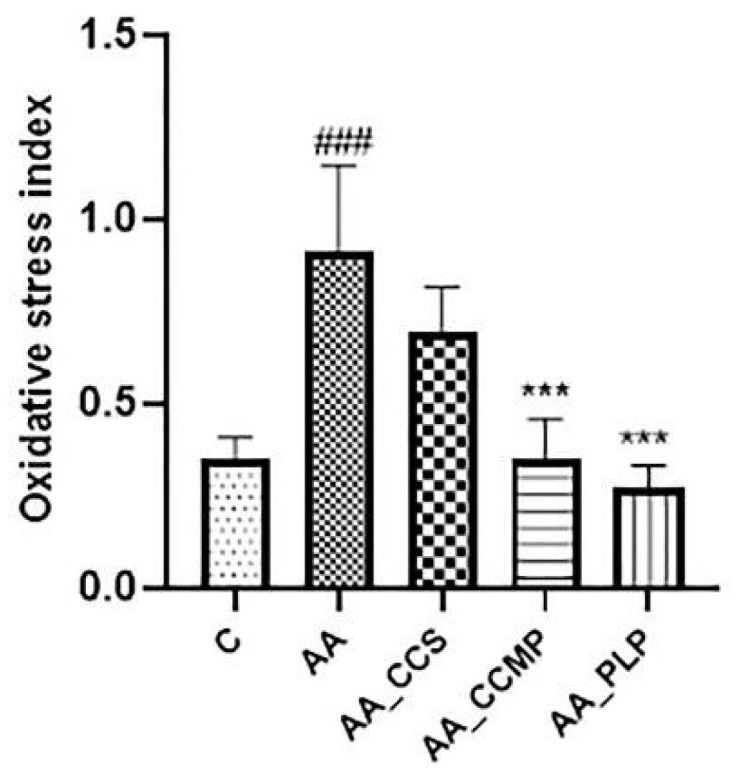
Oxidative stress index. Data are expressed as the mean ± SD for each group. ### *p* < 0.0001 indicates a significant difference in comparison with the control group; *** *p* < 0.0001 indicates a significant difference in comparison with the AA group. Abbreviations: C, normal control group; AA, acetic acid group; AA_CCS, curcumin group; AA_CCMP, Col-CUR-MPs group; and AA_PLP, prednisolone group. The bars represent mean values with standard deviations.

**Figure 4 molecules-27-05680-f004:**
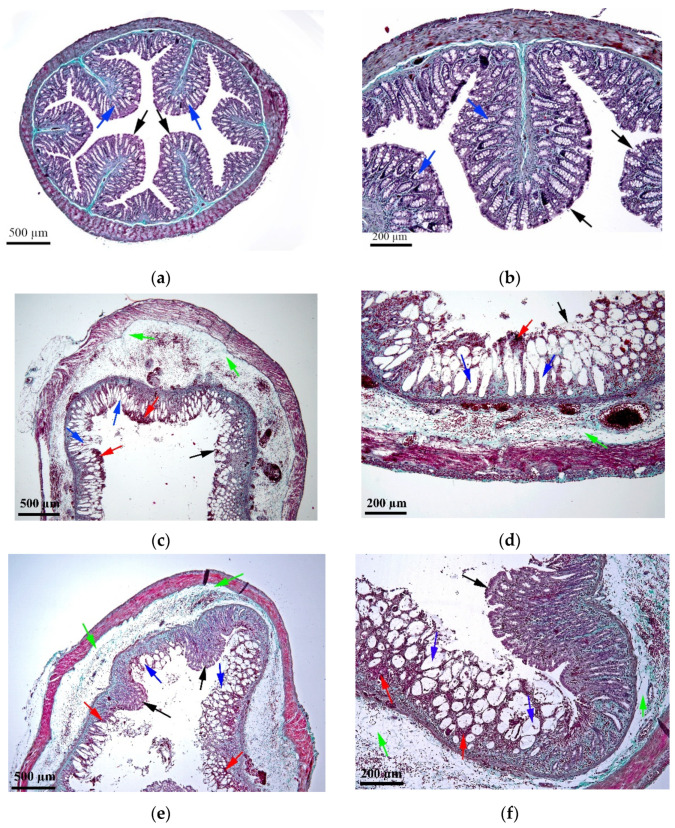
Representative images of colon tissues obtained from the rats included in the in vivo study—histopathological analysis, two images per group. (**a**,**b**) C, normal control group, overview and detail, respectively—showed a normal mucosa with preserved mucosal folds (black arrows), well-organized crypts with goblet cells (blue arrows); (**c**,**d**) AA, acetic acid group, overview and detail, respectively—showed generalized hemorrhagic necrosis (red arrows), severe edema (green arrows), altered mucosal folds (black arrows), loss of crypts with goblet cells (blue arrows); (**e**,**f**) AA_CCS, curcumin group, overview and detail, respectively—showed zonal hemorrhagic necrosis (red arrows), discrete submucosal edema (green arrows), partially preserved mucosal folds (black arrows), loss of crypts with goblet cells (blue arrows); (**g**,**h**) AA_CCMP, Col-CUR-MPs group, overview and detail, respectively—showed mild congestion (red arrows), discrete submucosal edema (green arrows), preserved mucosal folds (black arrows) and partially preserved crypts with goblet cells (blue arrows); (**i**,**j**) AA_PLP, prednisolone group, overview and detail, respectively—showed mild congestion (red arrows), discrete submucosal edema (green arrows), partially preserved mucosal folds (black arrows), loss of crypts with goblet cells (blue arrows). The scale bars represent 500 μm for the images which illustrate the cross-sections’ overview (**a**,**c**,**e**,**g**,**i**), while for the images which illustrate the details of sections (**b**,**d**,**f**,**h**,**j**), the scale bars represent 200 μm.

**Figure 5 molecules-27-05680-f005:**
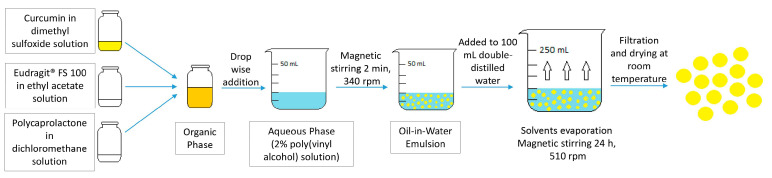
Col-CUR-MPs preparation method.

## Data Availability

Not applicable.

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
