# Peer review of "Curcumin-Loaded Microspheres Are Effective in Preventing Oxidative Stress and Intestinal Inflammatory Abnormalities in Experimental Ulcerative Colitis in Rats"

_molecules, 2022, doi:10.3390/molecules27175680_

Round 1
Reviewer 1 Report
This work investigates the effects of a novel curcumin-loaded polymeric microparticulate oral drug delivery system for colon targeting in an experimental model of ulcerative colitis. The work contains relevant information for the area and is congruent with the "Molecules" journal's scope. It deserves to be published after minor revision and refinement of some points:
1. From the title and the aim, the work commits to the study of "microspheres". However, the study does not include the characterization of the micro size and the spheric morphology of particles. It is recommended to complete analyses of the report's size and morphology of the particles obtained.
2. Page 10, lines 397-402. Please provide more details about the microparticle preparation process: concentration ratio of solutions/compounds, purification method, etc.
3. Page 3, line 137 (Results). I recommend starting this section by discussing something related to the preparation of the microspheres: size, morphology obtained, yield, etc. If this information is already published in the previous study, it is recommended to cite it in the materials and methods section.
Reviewer 2 Report
Hales et al. investigated very interesting study titled “Curcumin-loaded microspheres are successful in preventing oxidative stress and intestinal inflammatory changes in experimental ulcerative colitis”. It must, however, go through follow revisions before being considered for publication in the Molecules Journal.
1. In the title Successful is not a word that should be used. Therefore, I advise authors to alter the title to “Curcumin-loaded microspheres are effective in preventing oxidative stress and intestinal inflammatory abnormalities in experimental ulcerative colitis in rats”.
2. Abstract: A short outline of the current study's research gaps in relation to curcumin is required. Why did the authors decide to explore curcumin? This information should come before the objective. Begin on line 27 with “Male Wistar rats (n=40) were divided into five groups (n=8)……..Conclusion part need to be extended with future perspective.
3. Introduction: Reference(s) needed for lines 42–43 of the introduction. Additionally, because Paragraphs 2, 3, and 5 (Lines 58–88, 111–136) are too lengthy, I am no longer paying attention to them. It must be shortened. Lines 89–92 should be supported by the most recent reference (https://www.ncbi.nlm.nih.gov/pmc/articles/PMC8990857/).
4. Results: The result section is well written with clear Figures. Is it p<0.001 or p<0.0001? It is necessary to use arrows in Figure 4 to denote the alterations in the generalised necrotic-hemorrhagic enteritis and the severe edema.
5. Discussion: The discussion part is once again extremely lengthy and will undoubtedly lose readers' interest. Instead of adding more literature details, the author should offer critical justifications for the results that were found.
6. Materials and Methods: It is too long to explain regents. There is no need to include all of the common substances (For example Hexane, NaOH, HCl etc.). Just a few extremely specific substances must be mentioned. The remaining chemicals all simply state, "All other chemicals were of analytical grade." If at all possible, authors should include a schematic illustration or a graphic to clarify how the curcumin-loaded microspheres were prepared in Section 4.2. Because this is a novel idea, readers will be intrigued to learn about the process in a straightforward manner.
7. Conclusions: The findings/insights should be used to support this section. Finally, it will give a clear understanding of the study. Future viewpoints need to be covered in the conclusion. The importance of the study should be emphasised by the author.
Round 2
Reviewer 2 Report
I really appreciate the authors' effort because they made significant changes to the manuscript in response to the reviewer's suggestions. As a result, I recommend considering it for publishing in the Molecules Journal.